# Pt-Based Nanostructures for Electrochemical Oxidation of CO: Unveiling the Effect of Shapes and Electrolytes

**DOI:** 10.3390/ijms232315034

**Published:** 2022-11-30

**Authors:** Ahmed Abdelgawad, Belal Salah, Kamel Eid, Aboubakr M. Abdullah, Rashid S. Al-Hajri, Mohammed Al-Abri, Mohammad K. Hassan, Leena A. Al-Sulaiti, Doniyorbek Ahmadaliev, Kenneth I. Ozoemena

**Affiliations:** 1Center for Advanced Materials, Qatar University, Doha 2713, Qatar; 2Gas Processing Center, College of Engineering, Qatar University, Doha 2713, Qatar; 3Molecular Sciences Institute, School of Chemistry, University of the Witwatersrand, Private Bag 3, P O Wits, Johannesburg 2050, South Africa; 4Petroleum and Chemical Engineering Department, Sultan Qaboos University, Muscat 123, Oman; 5Nanotechnology Research Centre, Sultan Qaboos University, P.O. Box 17, PC 123, SQU, Al-Khoudh 123, Oman; 6Department of Petroleum and Chemical Engineering, College of Engineering, Sultan Qaboos University, P.O. Box 33, PC 123, SQU, A-Khoudh 123, Oman; 7Department of Mathematics, Statistics, and Physics, Qatar University, Doha 2713, Qatar; 8Andijan State Pedagogical Institute, Andijan 170100, Uzbekistan; 9Presidential School in Andijan, Agency for Presidential Educational Institutions of the Republic of Uzbekistan, Andijan 170100, Uzbekistan

**Keywords:** Pt porous nanodendrites, CO oxidation, effect of shape, effect of size, electrolyte effect

## Abstract

Direct alcohol fuel cells are deemed as green and sustainable energy resources; however, CO-poisoning of Pt-based catalysts is a critical barrier to their commercialization. Thus, investigation of the electrochemical CO oxidation activity (CO_Oxid_) of Pt-based catalyst over pH ranges as a function of Pt-shape is necessary and is not yet reported. Herein, porous Pt nanodendrites (Pt NDs) were synthesized via the ultrasonic irradiation method, and its CO oxidation performance was benchmarked in different electrolytes relative to 1-D Pt chains nanostructure (Pt NCs) and commercial Pt/C catalyst under the same condition. This is a trial to confirm the effect of the size and shape of Pt as well as the pH of electrolytes on the CO_Oxid_. The CO_Oxid_ activity and durability of Pt NDs are substantially superior to Pt NCs and Pt/C in HClO_4_, KOH, and NaHCO_3_ electrolytes, respectively, owing to the porous branched structure with a high surface area, which maximizes Pt utilization. Notably, the CO_Oxid_ performance of Pt NPs in HClO_4_ is higher than that in NaHCO_3_, and KOH under the same reaction conditions. This study may open the way for understanding the CO_Oxid_ activities of Pt-based catalysts and avoiding CO-poisoning in fuel cells.

## 1. Introduction

A worldwide squeeze on traditional non-renewable energy resources not only generated crippling deficiencies and surging fuel prices but also increased the carbon footprint [1,2,3,4]. The promising solutions are finding renewable energy resources [5,6,7,8,9,10] and gas conversion to high value-added products [1,2,3,4]. Electrochemical energy conversion and production technologies, such as alcohol fuel cells are considered eco-friendly, efficient, and sustainable energy sources due to their zero-emissions, outstanding power density, earth-abundance of alcohols, and low-operation temperature [11,12,13]. However, they remain impractical commerciality, especially for transportation, due to the high cost and CO-poisoning of Pt-based catalysts that is still the most active anode. Therefore, the CO oxidation (CO_Oxid_) reaction is important in multidisciplinary applications, such as industrial, fuel cells and environmental applications [1,2,14,15]. Various solutions were developed for improvement CO_Oxid_ of Pt-based catalysts focused on altering the shape (i.e., dimensions, porosity, and surface features), and composition (i.e., mixing with other metals) [16,17] to tailor the adsorption/activation of reactants (i.e., CO and O_2_) along with promoting dissociation of O_2_ and producing active oxygenated species (i.e., O• and •OH) needed for CO_Oxid_ under low voltage results is endowing. For instance, PtNi multicubes enhanced the CO_Oxid_ current density by 1.2 times and decreased the oxidation potential by 0.11 V than Pt/C in 1 M KOH electrolyte [18]. Pt_67_Bi_33_ nanosponge enhanced the CO_Oxid_ activity at a lower onset potential (0.52 V) compared to Pt/C (0.59 V) and Pt nanosponge (0.6 V) in 1 MHClO_4_ electrolyte [19]. The CO_Oxid_ current density of PtPd/CNs nanorods (14.75 mA cm^−2^) was superior to Pt/C (7.32 mA cm^−2^) by 2.01 time and bare CNs (0.63 mA cm^−2^) by 23.41 time beside a lower oxidation/onset potential in 0.1 M KOH electrolyte [20]. Another solution is to use a supporting material, such as carbon-based materials (i.e., carbon nanotubes, carbon nitride [20,21,22,23], and metal-organic framework [24,25], supported metals nanoparticles and transition metal oxides (i.e., TiO_2_, Pd/Ti_3_C_2_, PdCu/CN, Fe_2_O_3_, and Ce_2_O_3_) [26,27,28,29])), can enhance the CO_Oxid_ performance of Pt-based catalysts [17,30,31,32,33,34,35,36,37,38,39]. This is attributed to the enhanced electrical conductivity, abundant active sites, and electronic interaction with support, which strengthens the adsorption of reactants and facilitate the desorption of intermediates and products [40,41,42,43,44]. For instance, recently, we reported that the CO_Oxid_ activity of Pd NPs is enhanced significantly using Ti_3_C_2_T_x_ support [45]. However, the high tendency of Pt to aggregate and detach from the support during the catalytic reaction is a daunting challenge for the achievement of long-term durability.

Self-standing porous Pt-based nanostructures, especially nanodendrites, are highly promising for the CO_Oxid_ and other electrocatalytic applications. This is due to the outstanding surface, low density, rich edges, and interior/exterior cavities of Pt-based nanodendrites, which can accommodate guest species and ease their diffusion to the stable interior core resulting in maximizing utilization of surface/bulk Pt atoms alongside accelerating mass transfer [7,46,47,48]. For instance, Pd_52_Pt_48_ nanodendrites promoted the CO_Oxid_ activity with a lower onset/oxidation potential (0.45/0.823 V) than commercial Pt/C (0.507/0.831 V) in 0.1 M HClO_4_ electrolyte [49]. PtPdRu NDs showed a CO_Oxid_ density (10 mA/cm^2^) than PtPdRu flower (7.2 mAcm^2^), Pt/C (5.2 mA/cm^2^) and PtPd NDs (4.9 mAcm^2^) in 1 M NaOH solution [50]. Although the significant progress in the controlled synthesis of Pt-based catalysts for various electrocatalytic applications, the effect of their shape and electrolyte pH on the CO_Oxid_ is not yet emphasized died as far as we have found.

This contemplates us to the synthesis of porous Pt nanodendrites (Pt NDs) (10 ± 1 nm) and benchmark its CO_Oxid_ was compared to spherical Pt NPs (15 ± 2 nm) and commercial Pt/C NPs (4 ± 1 nm) catalysts in different electrolytes (i.e., NaHCO_3_, HClO_4_, and KOH) under various pH range. Pt NDs were synthesized via ultrasonic irradiation at room temperature in the presence of polyvinylpyrrolidone as a morphology-directing agent, while Pt spherical nanoparticles (Pt NPs) were formed via the chemical reduction method. For the first time, the effect of Pt shape and size, in addition to the pH of the electrolytes (i.e., acidic, neutral, and alkaline), on the CO_Oxid_ activity and durability are investigated using electrochemical techniques at room temperature.

## 2. Results and Discussion

Figure 1a shows the fabrication process of Pt NDs formed via chemical reduction of Pt-slat by AA in the presence of PVP as a structure-directing agent, based on the acoustic cavitation mechanism of ultrasonic waves that isolate nucleation from the growth step [50]. Meanwhile, using NaBH_4_ without sonication produces chains-like nanostructures Pt NCs, based on direct nucleation and growth due to the strong reduction power of NaBH_4_ (Figure 1a) [19]. Pt NDs were formed in a high-yield (~100%) of uniform spatial porous nanodendrites (Figure 1b) with an average size of (10 ± 1) nm (Figure 1c) and abundant pores as indicated by the circles in (Figure 1d). The HRTEM image of individual particle reveals that NDs comprises multiple arms with an average diameter of (2 ± 0.3 nm) assembled in a three-dimensional branched morphology with various cavities in the inner and outer area as indicated by the arrows in (Figure 1e). The lattice fringes are uniform without any crystal defects and are coherently extended from the inner core to the branches in a different direction, implying the non-epitaxial growth that could be severe as an indication of homogenous nucleation and subsequent growth rather than random agglomeration (Figure 1e) [50]. This is further seen in the Fourier filtered HRTEM images, which indicate the lattice fringes attributed to (111), (200), and (220) facets of face-centered-cubic (fcc) of Pt, as generally observed in Pt-based catalysts (Figure 1f–h).

These results are in line with the selected area electron diffraction (SAED) patterns that display the typical rings assigned to infer the typical diffraction rings assigned to (111), (200), (220), and (311) planes of fcc Pt (Figure 1i). The HAADF-STEM depicts the porous dendritic shape with abundant interior and exterior cavities among arms (Figure 1j). The TEM image of Pt NCs reveals the formation of nanochains-like nanostructures with an average size of (6 ± 0.5 nm in width) (Figure 1k–m). The HRTEM image of Pt NCs shows the lattice fringes assigned to (111) and (200) facets of fcc Pt (Figure 1n–p), meanwhile the SAED indicates the main facets of fcc Pt (Figure 1q). The commercial Pt/C catalyst have semi-spherical nanoparticles with an average diameter of (4 ± 0.3 nm) distributed over carbon sheets (Appendix A).

Figure 2a shows the XRD diffractions pattern of Pt NDs relative to Pt NCs and commercial Pt/C, which all reveal the peaks corresponding to (111), (200), (220), and (331) facets of fcc Pt in line with (JCPDS:04-0802) [11,51]. Pt/C showed an additional broad peak at 2*Ɵ* of 25° attributed to the (002) facet of amorphous graphitic carbon. The absence of any Pt-O peaks is indicative of the uniformity of the prepared catalysts without undesired crystalline phases, as usually observed in Pt-based nanostructures formed using reducing agents. The diffraction peaks of Pt NDs are less intense and with more broaden compared with Pt NCs and Pt/C due to the difference in the crystallite size. Thereby, the crystallite size of Pt NDs (3.5 nm) is slightly larger than that of Pt NCs (3 nm) and Pt/C (2 nm) calculated by the Scherrer equation from the full width at half maximum of the (111) diffraction peak.

Figure 2b shows the XPS survey of Pt NDs, Pt NCs, and Pt/C, which all display core levels of the Pt 4f and C 1s. The high-resolution XPS spectra of Pt 4f in all catalysts show Pt 4f_7/2_ and Pt 4f_5/2_, but with a noticed slight positive shift in the binding energies of Pt 4f in Pt NDs and Pt NCs compared to Pt/C, possibly attributed to decreasing the d-band center of Pt (Figure 2c). Alteration of the d-band of Pt with respect to the fermi level allows tuning the adsorption energies of reactants besides the desorption of intermediates and products during the electrocatalytic reactions. The fitting of Pt 4f spectra of Pt NDs depicts (4f_7/2_ and 4f_5/2_) assigned to Pt^o^ with an atomic percentage of (80.1%) as the main phase and a small plateau assigned to Pt^2+^ (Figure 2d), implying the existence of Pt in Pt^o^ metallic phase. This originated from the high reduction power of ascorbic acid towards Pt precursors under sonication, as reported elsewhere [19,50,51,52]. Appendix A shows the binding energies of Pt 4f phases.

Figure 3a depicts CV curves of Pt NDs, Pt NCs, and commercial Pt/C measured N_2_-saturated an aqueous solution of M HClO_4_ electrolyte, which all reveal the main voltammogram features of Pt, including hydrogen under-potential adsorption/desorption (H-UPD) at (−0.25 to +0.25 V), a double layer at (0.2–0.45 V), and Pt-H/Pt-O at higher potentials. Both Pt NDs and Pt NCs show more characteristic Pt-H/Pt-O than Pt/C, implying their ease of formation oxide and their reduction, plausibly due to their ability to weaken and delay formation of Pt-oxygenated species as inferred in the positive shift in redox waves of Pt NDs and Pt NCs than Pt/C. Notably, Pt NDs and Pt NCs show a greater H-UPD area than Pt/C, suggesting their abundant active sites and higher active surface area. Thereby, the ECSA_H-UPD_ of Pt NDs (59.1 m^2^/g) is higher than that of Pt NCs (52.2 m^2^/g) and Pt/C (48.3 m^2^/g) (Appendix A). This is due to the unique surface characteristics of 3D porous nanodendritic and 1D nanochain morphologies with high aspect ratios, which are imminent with their great surface area relative to 0D nanospheres.

Figure 3b shows the CVs of Pt NDs, Pt NCs, and Pt/C tested in CO-saturated HClO_4_ electrolyte, displaying the CO_Oxid_ voltammogram features including a sharp oxidation peak in the forward direction and a weak reduction peak in the backward direction but with a superior activity of Pt NDs than Pt NCs and Pt/C. It can be seen that the reduction currents in N_2_ (Figure 3a) and CO (Figure 3b) saturated electrolytes are not the same, due to the strong binding of CO to Pt results in a low surface coverage of OH which not decrease the accessible active sites but also reduce the oxidative removal of CO oxidation intermediates species and Pt-reduction. This leads to difference between the magnitude of both reduction currents as noticed before during CO oxidation on Pt-based or Pd-based catalysts [24,25,53,54]. The integrated CO oxidation charges from CV curves of Pt NDs, Pt NCs, and Pt/C in the forward scan are approximately 359.5, 160, and 151 μC, respectively, implying the superior CO_ads_ ability of Pt NDs and Pt NCs than Pt/C. So, the ECSA extracted by the underpotential adsorption of CO (ECSA_CO-UPD_) of Pt NDs (34.23 m^2^/g) is higher than that of Pt NCs (15.23 m^2^/g), and Pt/C (14.38 m^2^/g), imply the grater accessible active sites in Pt NDs (Appendix A), which is in line trend with the ECSA_H-UPD_. This is evidenced in the higher CO_Oxid_ current density in the forward direction (*I_f_*) of Pt NDs (2.03 mAcm^−2^) by 2.18 time than Pt NCs (0.93 mAcm^−2^) and by 3.44 than Pt/C (0.59 mAcm^−2^), due to the abundant active sites and porous morphology which maximize utilization of buried Pt atoms. This is shown in the ability of Pt NDs and Pt NCs to preserve their H-UPD during CO_Oxid_, as indicated by the arrows in (Figure 3b). The ECSAs obtained from integration of the CO oxidation peak (Appendix A) seem to show an increasing trend going from Pt/C to Pt NCs to Pt NDs. The CO_Oxid_ current (I*_f_*) of Pt NDs (2.03 mAcm^−2^) is greater than that of previously reported Pt-based catalysts, such as PtRu@h-BN/C [55], Pt dendrimer-encapsulated nanoparticles [56], Pt-NbOx [57], and Pt/C [58] measured under similar conditions (Appendix A). The electrochemical CO oxidation current densities were normalized comparatively by the geometric area of the working electrode *j_geo_*) and to the ECSA (*j_ECSA_*), revealing the higher activity of Pt NDs than Pt NCs and Pt/C catalyst (Appendix A). The estimated *j_geo_*/*j_ECSA_* are approximately Pt NDs 2.03/0.027, 0.93/0.013, and 0.59/0.008 mA/cm^2^, respectively. This implies the superior intrinsic electrochemical CO oxidation activity of Pt NDs than Pt NCs and Pt/C catalysts at the same Pt-loading. The CO_Oxid_ potential (*E_Oxid_*) on Pt NDs (0.45) is 0.1 V and 0.2 V lower than that on Pt NCs, and Pt/C; respectively (Figure 3c), implying the ease of desorption of intermediates and byproducts, which in turn accelerate the CO_Oxid_ kinetics on Pt NDs. This is seen in the earlier onset potential (*E_Ons_*) (0.33 V) on Pt NDs than Pt NCs (0.36 V) and Pt/C (0.58 V). Thereby, Pt NDs deliver a greater I*_f_* under any applied potential point than Pt NCs and Pt/C as indicated by the dashed lines in (Figure 3c), that serve as evidence for the faster CO_Oxid_ kinetics of Pt NDs. The CVs curves are recorded at different scan rates on the as-synthesized catalysts to get more shades of the CO_Oxid_ process. The I*_f_* increases with increasing the scan rates to reach the highest value at 200 mV/s, but with an obvious superiority of Pt NDs (Figure 3d) over Pt NCs (Figure 3f) and Pt/C (Figure 3h). The current density for all samples was plotted against the square root of the scan rate, and all tested catalysts reveal a linear relation that plausibly indicates the CO_Oxid_ is a diffusion-controlled process. The line slope for Pt NDs (0.65) (Figure 3e) is greater than that of Pt NCs (0.47) (Figure 3g) and Pt/C (0.4) (Figure 3i), inferring the better transportation kinetics on Pt NDs and Pt NCs than Pt/C, owing to the shape effect [20].

The CA measurements show greater durability of Pt NDs than Pt NCs and Pt/C over 2000 s (Figure 4a). The CV curves measured after chronoamperometry (CA) tests show that all samples maintain their initial CO_Oxid_ voltammogram features but with higher stability on Pt NDs than Pt NCs and Pt/C (Figure 4b–d). Mainly, the I*_f_* of Pt NDs degraded only by 20% compared with Pt NCs (22%) and Pt/C (27%) due to the morphology effect, which maintains higher ECSA and preserves active sites from blocking. Mainly, self-standing Pt NDs and Pt NCs shapes are not susceptible for aggregation compared to supported Pt/C nanosphere. The ECSA of Pt NDs is almost maintained with only 10% degradation relative to Pt NCs (19%) and Pt/C 29%) (Figure 4f).

This is corroborated in the TEM images recorded after durability tests, which show the significant morphological stability of Pt NDs and Pt NCs, while Pt/C shows detachment and aggregation of Pt nanoparticles (Appendix A). The Nyquist plots of EIS tests imply the semicircle lines but with a smaller diameter for Pt NDs than PtNCs and Pt/C, which serve as an evidence for the morphology effect on increasing the charge transfer across the electrolyte–electrode interface (Figure 4e). This is confirmed by the fitting and analysis of EIS data based on the Voigt electrical equivalent circuit (EEC), which shows lower electrolyte resistance (*R*_s_) and charge transfer resistance (*R*_ct_) of Pt NDs than Pt NCs and Pt/C (Table 1). In addition, Pt NDs have greater constant phase elements (CPE) than Pt NCs and Pt/C catalysts, implying its faster charge mobility. The variation in the R_s_ is plausibly due to the morphology, active sites, and surface charges on the catalysts along with their dissimilar interaction with the electrolyte. Another possible factor is testing the EIS at the CO oxidation potential of each catalyst, which are all different, so the charge transfer across the electrolyte–electrode interface of each catalyst will not be similar and hence the obtained R_s_ will be different. This is shown in the EIS results, which revealed semicircle lines, but with a smaller diameter for Pt NDs than Pt NCs and Pt/C, due to the morphology effect *(*Figure 4e) in line with elsewhere reports [24,25].

The CO_Oxid_ mechanism could be proposed according to Langmuir–Hinshelwood (Equations (1)–(3)):(1)$$\mathrm{Pt}\;\mathrm{NDs}+\mathrm{CO}\;\to \;\mathrm{Pt}\;\mathrm{NDs}\text{-}{\mathrm{CO}}_{\mathrm{ads}}$$
(2)$$\mathrm{Pt}\;\mathrm{NDs}+H_{2}O\;\to \;\mathrm{Pt}\;\mathrm{NDs}\text{-}{\mathrm{OH}}_{\mathrm{ads}}+H^{+}+e^{-}$$
(3)$$\mathrm{Pt}\;\mathrm{NDs}\text{-}{\mathrm{OH}}_{\mathrm{ads}}+\mathrm{Pt}\;\mathrm{NDs}\text{-}{\mathrm{CO}}_{\mathrm{ads}}\;\to \;{\mathrm{CO}}_{2}+2\mathrm{Pt}\;\mathrm{NDs}+H^{+}+e^{-}$$

Which includes the co-adsorption of CO (CO_ads_) and hydroxyl OH adsorption (OH_ads_) on the Pt surface. Notably, OH is generated from water dissociation (H_2_O ↔ H^+^ + OH^−^) in the electrolyte using CO-free sites of Pt [19]. Then, OH_ads_ facilitate CO_Oxid_ to produce CO_2_ that is subsequently desorbed from the Pt surface. Therefore, it seems that Pt NDs and Pt NCs are able to tolerate the CO_ads_ on some Pt active sites besides promoting the generation of OH species, and its adsorption on CO-free sites results in great CO_Oxid_ activity and durability [19].

The CO_Oxid_ is conducted in KOH and NaHCO_3_ electrolytes to investigate the effect of electrolyte pH on the CO_Oxid_. The CV curves tested in N_2_-pursed KOH on the Pt NDs, Pt NCs, and Pt/C catalysts show only Pt voltammogram. Thereby, the ECSA_H-UPD_ of Pt NDs (66 m^2^/g) and Pt NCs (65.2 m^2^/g) are higher than that of Pt/C (52.5 m^2^/g) (Appendix A). Meanwhile, after CO pursing, the CO_Oxid_ voltammogram appeared but was slightly different from those measured in HclO_4_ electrolyte, including minute peak with multiple oxidation peaks in the forward and reduction peaks in the backward scan in line with elsewhere reports (Figure 5a). The integrated CO oxidation charges from CV curves of Pt NDs, Pt NCs, and Pt/C in the forward scan are approximately 322.5, 232.9, and 192 μC, respectively, implying the superior CO_ads_ ability of Pt NDs and Pt NCs than Pt/C. So, the ECSA_CO-UPD_ of Pt NDs (30.71 m^2^/g) is greater than that of Pt NCs (22.18 m^2^/g) and commercial Pt/C catalyst (18.28 m^2^/g) (Appendix A). The dissimilar oxidation behaviors are attributed to dissimilar abilities of anions adsorption from the electrolyte and water dissociation allowing OH_ads_ onto active sites of Pt while in the acidic electrolyte, OH_ads_ is excluded from active sites (i.e., the dipole moment at defect/step sites, which are inherently attractive to anions). The I*_f_* anodic current on Pt NDs (1.71 mA/cm^2^) is 1.54 time than Pt NCs (1.11 mA/cm^2^) and 3.28 time than Pt/C (0.52 mA/cm^2^), implying the maximized utilization of Pt in NCs and NDs, due to their accessible active sites (Figure 5b). The *j*_ECSA_ of Pt NDs (0.078 mA/cm^2^) is also higher than that of Pt NDs than Pt NCs(0.07 mA/cm^2^) and Pt/C (0.04 mA/cm^2^) catalysts, indicating its higher intrinsic electrochemical CO oxidation activity (Appendix A). The E_ons_ on Pt NDs (−0.5V) are lower than that on Pt NCs (−0.48 V) and Pt/C (−0.43), implying the faster CO_Oxid_ kinetics on Pt NDs. The increment in the *I_f_* with increasing scan rates and their linear relationship on catalysts may indicate that the CO_Oxid_ is a diffusion-controlled process (Figure 5c,e,g). Pt NDs show a larger line slope (0.28) than Pt NCs (0.23) and Pt/C (0.14) (Figure 5d,f,h), demonstrating the better transportation kinetics on Pt NDs and Pt NCs than Pt/C, resulting from the shape effect [20].

The CA curve of Pt NDs reveals a slower current attenuation with higher current retention after 2000 s than that of Pt NCs and Pt/C (Figure 6a). The CV curves after CA tests show the degradation of I_f_ anodic current on Pt NDs by (19%) relative to Pt NCs (22%) and Pt/C (30%), implying the greater durability of Pt NDs (Figure 6b–d).

The EIS measurements imply a lower charge transfer resistance on Pt NCs and Pt NDs than Pt/C, indicating the effect of morphology, which enhances the charge transfer and facilitates the electrolyte-electrode interaction (Figure 6e). This is proved by fitting the EIS data, which reveals the lower *R*_s_ and *R*_ct_ along with a higher CPE of Pt NDs than Pt NCs and Pt/C (Table 2). So, the ECSA of Pt NDs is almost maintained with only 11% degradation relative to Pt NCs (29%) and Pt/C 42%) (Figure 6f).

Unlike the CV curves measured in N_2_-saturated NaHCO_3_ electrolyte (0.5 M), the CV curves measured under continuous CO purging depict the CO_Oxid_ voltammogram characteristics with a higher activity of Pt NDs and Pt NCs than Pt/C as noticed in KOH and HClO_4_ electrolytes (Figure 7a). Particularly, the integrated CO oxidation charges from CV curves of Pt NDs, Pt NCs, and Pt/C in the forward scan are approximately 205, 198, and 118 μC, respectively, implying the superior CO_ads_ ability of Pt NDs and Pt NCs than Pt/C. Thereby, the ECSA_CO-UPD_ of Pt NDs (19.52 m^2^/g) and Pt NCs (18.85 m^2^/g) are higher than that of Pt/C (11.23 m^2^/g). We could not calculate the ECSA_H-UPD_, due to the unclear H-UPD area. The *j*_geo_ of Pt NDs (0.5 mA/cm^2^) is greater than Pt NCs (0.47 mA/cm^2^) and Pt/C (0.23 mA/cm^2^). The greater intrinsic electrochemical CO oxidation activity of Pt NDs is manifested in its higher *j*_ECSA_ (0.036 mA/cm^2^) than that of Pt NCs (0.035 mA/cm^2^) and Pt/C (0.028 mA/cm^2^) at equivalent Pt mass (Appendix A). Additionally, the E_Ons_ on Pt NDs (−0.11 V) are 0.04 V and 0.09 V lower than that on Pt NCs (−0.07 V) and Pt/C (−0.2), implying the accelerated CO_Oxid_ kinetics on Pt NDs and NCs, due to their ability to weaken the adsorption of intermediates and ease desorption of products under low potential results in quick regeneration of the active sites. The CV curves show the enhancement in the I*_f_* with increasing sweeping rates (Figure 7c,e,g), and their linear relationship on catalysts indicate that the CO_Oxid_ is a diffusion-controlled process similar to the results obtained in KOH and HClO_4_ electrolytes. The larger line slope of Pt NDs (0.16) compared to Pt NCs (0.15) and Pt/C (0.089), suggest the ease electron migration kinetics on Pt NDs and Pt NCs than Pt/C (Figure 7d, f, and h). Pt NDs are more stable than Pt NCs and Pt/C as seen in the CA curves measured for 2000 s (Figure 8a) and lower loss in the *I_f_* (15%) than Pt NCs (18%) and Pt/C (25%) (Figure 8b–d). Furthermore, Pt NDs and Pt NCs display better charge mobility than Pt/C, as shown in the lower charge transfer resistance obtained from EIS measurements (Figure 8e). This is verified by fitting the EIS data, which reveals the lower *R_s_* and *R*_ct_ along with a higher CPE of Pt NDs than Pt NCs and Pt/C (Table 3).

The ECSA of Pt NDs is almost maintained with only 19.5% degradation relative to Pt NCs (35%) and Pt/C (50%) (Figure 8f).

All in all, these results warrant the substantial effect of the morphologies of the catalysts and the pH of electrolytes on promoting the CO_Oxid_ activity and stability of Pt NDs, Pt NCs, and Pt/C catalysts. The self-standing Pt NDs outperformed Pt NCs and Pt/C due to the 3D porous structure, multiple surface corners, and interior cavities, which ease the adsorption of reactants and allow their diffusion to inner cavities that are stable and not feasible for aggregation [7,11,50]. This led to maximizing the utilization of buried Pt atoms in the core area, as noticed in the higher CO adsorption and I*_f_* of Pt NDs. Meanwhile, Pt NDs can facilitate water dissociation, which in turn allows the oxidative removal of intermediates and eases the CO_Oxid_ process at a lower potential. Pt NCs, with its 1D structure, high aspect ratio, and stable core, endowed the CO_Oxid_ more than Pt/C catalysts.

## 3. Methods and Materials 

### 3.1. Chemicals and Materials

Potassium tetrachloroplatinate(II) (K_2_PtCl_4_, ≥99.99%), L-ascorbic acid (≥99%), polyvinylpyrrolidone (Mwt 40000), and commercial Pt/C catalyst E-TEK Pt/C (20% Pt) were purchased from Sigma-Aldrich Chemie GmbH (Munich, Germany). Perchloric acid ((HClO_4_), ≥70%), potassium hydroxide (KOH ≥ 85%, pellets), and sodium bicarbonate ((NaHCO₃), MW: 84.01 g/mol) were obtained from (VWR Chemicals BDH).

### 3.2. Synthesis of porous Pt nanodendrites

Porous Pt nanodendrites (Pt NDs) were formed via mixing of an aqueous solution of K_2_PtCl_4_ (2m L of 10 mM), PVP (0.1 g) and ascorbic acid (0.5 mL, 0.4 M) under ultrasonic treatment (10 kW) for 10 min at room temperature. Then, the Pt NDs were isolated via 4 centrifugation cycles at 10k rpm for 10 min and washed with water (18.2 MΩ × cm resistance).

### 3.3. Synthesis of spherical Pt nanoparticles

Spherical-like Pt nanoparticles (Pt NPs) were synthesized via mixing of an aqueous solution of K_2_PtCl_4_ (2m L of 10 mM), PVP (0.1 g) and NaBH_4_, (1 mL of 10 mM) under magnetic stirring in the ice bath for 10 min. Then, the Pt NPs were separated via the four-centrifugation cycles at 10k rpm for 10 min and washed with water (18.2 MΩ × cm resistance).

### 3.4. Electrochemical CO_Oxid_ reaction

The electrocatalytic CO_Oxid_ performance was conducted using cyclic voltammogram (CVs), linear sweep voltammogram (LSV), impedance spectroscopy (EIS), and chronoamperometry (CA) tests on Gamry potentiostat (Reference 3000, Gamry Co., Warminster, PA, USA). A three-electrode cell comprised of Pt wire, Ag/AgCl, and glassy carbon ((⌀5 mm × 1 mm) as a counter, reference, and working electrodes, respectively, was used in all measurements. The glassy carbon electrodes were initially polished via 1 μm, 0.03 μm, and 0.05 μm alumina powder and then rinsed by ethanol and in deionized water 3 times under sonication for 3 sec. The catalyst inks (2 mg) were dissolved in an aqueous solution of 1 mL ethanol/H_2_O (3/1 *v*/*v*) under sonication using (P30H Ultrasonic water-Bath) under frequency of 80 kHz at room temperature for 10 min. Then the catalysts inks were deposited volumetrically onto the glassy-carbon electrode and then covered with 5 µL of Nafion^R^ solution (0.1 wt.%) and left to dry in an oven at 50 °C for 1 h. The catalysts loading on the working electrode was adjusted to (5 µg/cm^2^ of Pt) using the inductively coupled plasma optical emission spectrometry (ICP-OES, Agilent 5800). Prior to the measurements the CV test was measured in each electrolyte under continuous pursing of N_2_ at 200 mV/s for 100 cycles to remove any impurities and then measured at 50 mV/s for 3 cycles to measure the electrochemical active surface area (ECSA). The potential ranges are mainly determined based on the electrolytes including (−0.3 V to 1 V in 0.1 M HClO_4_), (0.3 V to −1 V in 0.1 M KOH), and (−0.5 V to 0.5 V in 0.5 M NaHCO_3_). After that, the electrodes were moved to another cell with fresh electrolytes under continuous CO pursing to measure the CO oxidation reaction. The ECSA was estimated by the hydrogen under-potential adsorption/desorption (H-UPD) (ECSA_H-UPD_) using this equation: ECSA_H-UPD_ = QH/m × 210, where Q_H_ is the charge for H-UPD obtained after the double layer correction area, m is the mass of the catalyst, and 210 µC/cm^2^ is the charge required for adsorption of a monolayer of hydrogen onto Pt surface. The same equation was also used for calculation of the ECSA using the underpotential adsorption of (CO-UPD) (ECSA_CO-UPD_), but QH is the integration of CO-UPD in the forward scan directions. The EIS measurements were carried out under the CO oxidation potential of each catalyst at frequency ranged from 0.1 Hz to 100 kHz with an AC voltage amplitude of 5 mV at open circuit potential in different electrolytes. The Voigt electrical equivalent circuit (EEC) was used for fitting and analysis the EIS measurements. For the bulk CO oxidation, the electrodes were initially cleaned in N_2_-saturated electrolyte solution via CV scans for several cycles and then immersed in a fresh electrolyte under CO pursing for 10 min and all the measurements were tested under CO pursing [53,54].

### 3.5. Materials Characterization

The transmission electron microscope (TEM) was conducted on (TEM, TecnaiG220, FEI, Hillsboro, OR, USA) equipped with high-angle annular dark-field scanning transmission electron microscopy (HAADF-SEM) and energy dispersive spectrometer (EDS). The X-ray photoelectron spectroscopy (XPS) spectra were recorded on a Kratos Axis (Ultra DLD XPS Kratos, Manchester, UK). The X-ray diffraction patterns (XRD) were recorded on an X-ray diffractometer (X’Pert-Pro MPD, PANalytical Co., Almelo, The Netherlands).

## 4. Conclusions

In brief, we have synthesized 3D porous Pt nanodendrites (Pt NDs) and 1D Pt nanochains (Pt NCs) through chemical reduction under sonication and systematically studied their CO_Oxid_ performance relative to commercial 0D Pt/C catalysts in HClO_4_, KOH, and NaHCO_3_ to emphasize the shape and electrolyte effect. The CO_Oxid_ activity and stability of Pt NDs is higher than that of Pt NCs and Pt/C catalysts in HClO_4_, KOH, and NaHCO_3_ electrolytes, respectively. This is shown in the garter ECSA, I_f_, and CO adsorption besides a lower onset potential and charge transfer resistance on Pt NDs than Pt NCs and Pt/C under the same condition owing to the 3D porous branched structure with a high surface area and accessible inner/outer active sites, which maximize Pt utilization. All catalysts reveal different CO_Oxid_ voltammogram features and kinetics in different electrolytes but with superior activities in HClO_4_ than that in KOH and NaHCO_3_. This study indicates that self-standing anisotropic Pt-based morphologies (i.e., 3D dendritic and 1D chain) are preferred over 0D Pt nanospheres supported on C, which may open the way for understanding the CO poisoning of Pt-based catalyst alcohol oxidation-based fuel cells. 

## Figures and Tables

**Figure 1 ijms-23-15034-f001:**
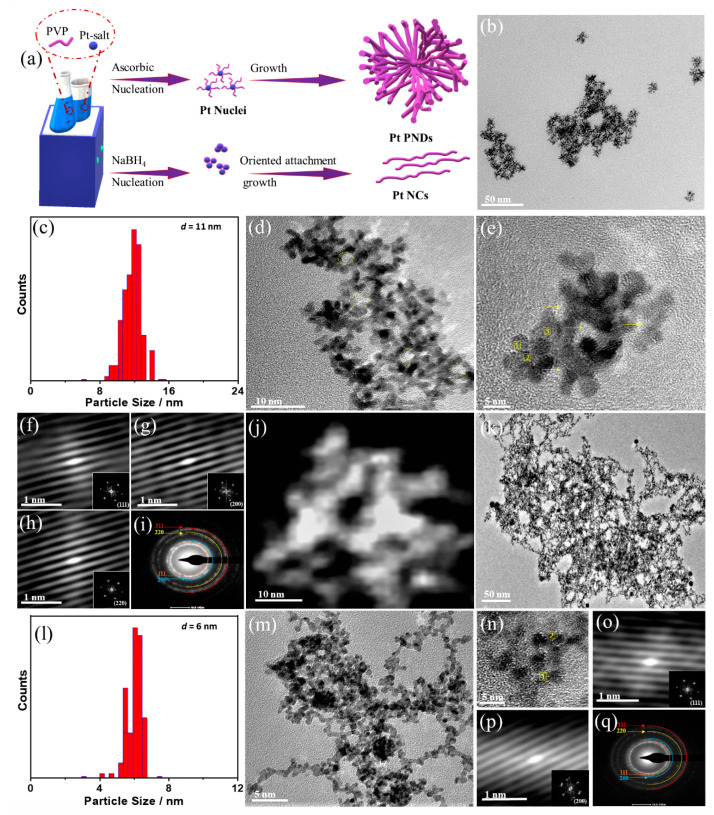
(**a**) Schematic illustration of the preparation process of Pt NDs and Pt NCs. (**b**) TEM image, (**c**) particle size distribution, (**d**) high-magnification TEM image, (**e**) HRTEM of Pt NDs, and (**f**–**h**) Fourier filtered HRTEM images of the numbered areas (1–3) in (**e**) respectively, (**i**) SAED, and (**j**) HAADF-STEM image of Pt NDs. (**k**) TEM images, (**l**) particle size distribution, (**m**) high-magnification TEM image, (**n**) HRTEM of Pt NCs and (**o**) SAED, and (**p**,**q**) Fourier filtered HRTEM images of the numbered areas (1–2) in (**n**), respectively, of Pt NDs.

**Figure 2 ijms-23-15034-f002:**
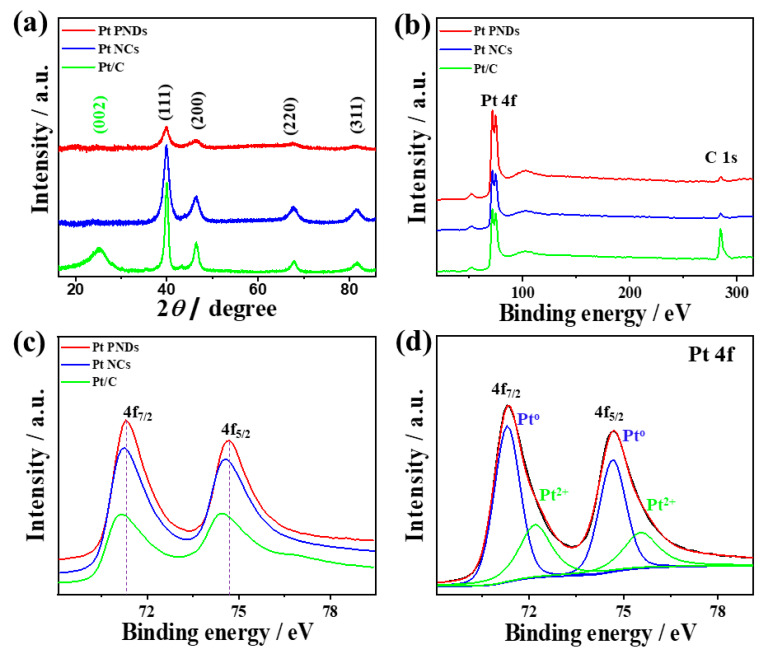
(**a**) XRD, (**b**) XPS survey (**c**) high-resolution spectra of Pt 4f in Pt NDs, Pt NCs, and Pt/C. (**d**) Fitting of Pt 4f spectra I Pt NDS.

**Figure 3 ijms-23-15034-f003:**
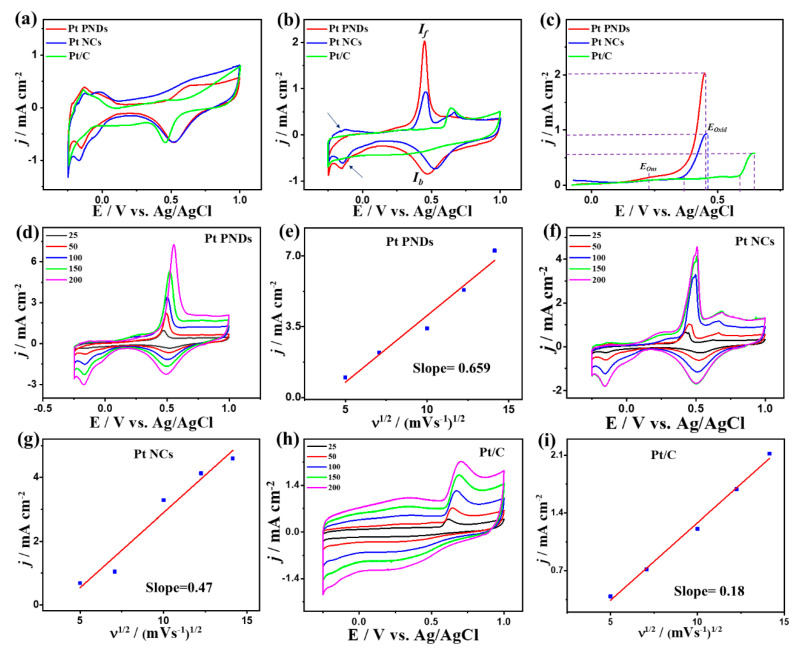
(**a**) CVs in N_2_-staruraed 0.1 M HClO_4_, (**b**) CO-saturated 0.1 M HClO_4_ at 50 mV/s, and (**c**) LSV at 50 mV/s of Pt NDs, Pt NCs, and Pt/C. CV curves at different scan rates and their related plots of *I_f_* vs. *v*^1/2^ on (**d**,**e**) Pt NDs, (**f**,**g**) Pt NCs, and (**h**,**i**) Pt/C in CO-saturated 0.1 M HClO_4_.

**Figure 4 ijms-23-15034-f004:**
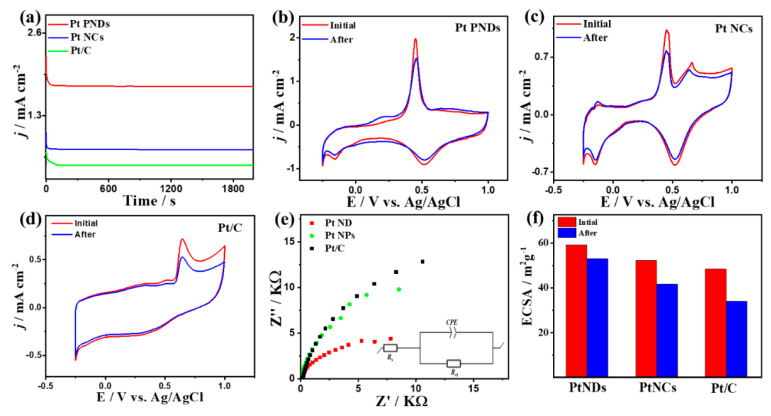
(**a**) CA tests measured on Pt NDs, Pt NCs, and Pt/C in CO-saturated 0.1 M HClO_4_. CV curves measured after CA on (**b**) Pt NDs, (**c**) Pt NCs, and (**d**) Pt/C. (**e**) EIS and (**f**) ECSA stability.

**Figure 5 ijms-23-15034-f005:**
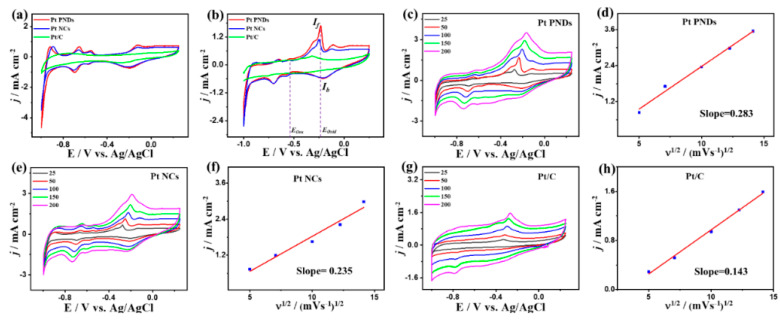
CVs in (**a**) N_2_-staruraed 0.1 M KOH, (**b**) in CO-saturated 0.1 M KOH at 50 mV/s on Pt NDs, Pt NCs, and Pt/C. CV curves at different scan rates and its related plots of *I_f_* vs. *v*^1/2^ on (**c**,**d**) Pt NDs, (**e**,**f**) Pt NCs, and (**g**,**h**) Pt/C in CO-saturated 0.1 M KOH.

**Figure 6 ijms-23-15034-f006:**
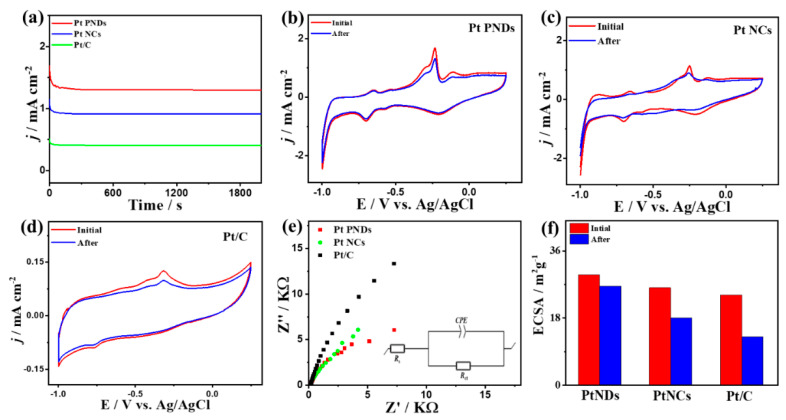
(**a**) CA tests measured on Pt NDs, Pt NCs, and Pt/C in CO-saturated 0.1 M KOH. CV curves measured after CA on (**b**) Pt NDs, (**c**) Pt NCs, and (**d**) Pt/C. (**e**) EIS and (**f**) ECSA stability.

**Figure 7 ijms-23-15034-f007:**
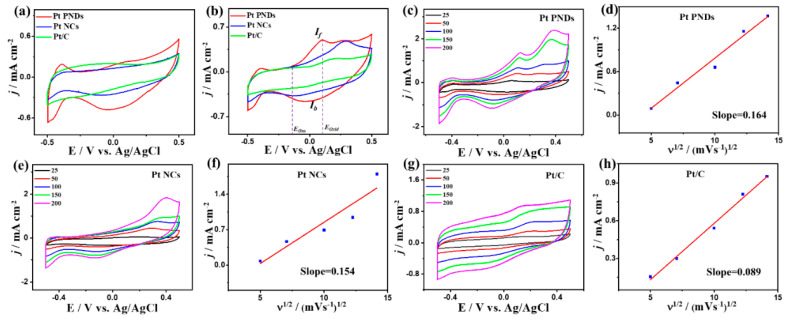
CVs in (**a**) N_2_-staruraed 0.5 M NaHCO_3_, (**b**) in CO-saturated 0.5 M NaHCO_3_ at 50 mV/s on Pt NDs, Pt NCs, and Pt/C. CV curves at different scan rates and its related plots of *j* vs. *v*^1/2^ on (**c**,**d**) Pt NDs, (**e**,**f**) Pt NCs, and (**g**,**h**) Pt/C in CO-saturated 0.1 M KOH.

**Figure 8 ijms-23-15034-f008:**
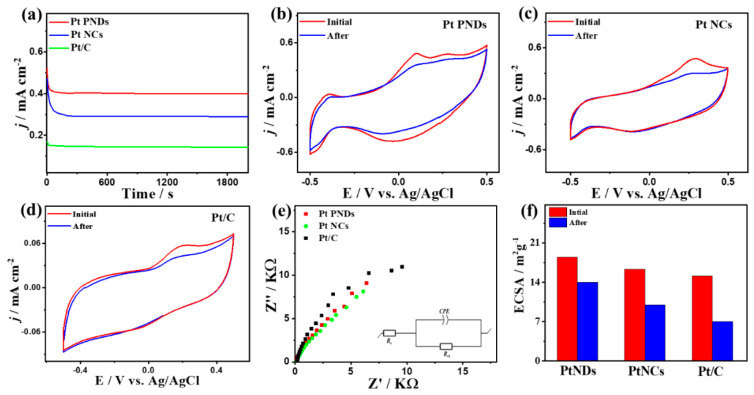
(**a**) CA tests measured on Pt NDs, Pt NCs, and Pt/C in CO-saturated 0.5 M NaHCO_3_. CV curves measured after CA on (**b**) Pt NDs, (**c**) Pt NCs, and (**d**) Pt/C. (**e**) EIS and (**f**) ECSA stability.

**Table 1 ijms-23-15034-t001:** Analysis of EIS measured in 0.1 M HClO_4_.

Catalyst	R_s_ (Ω)	R_ct_ (kΩ)	CPE (μF.s^(1−a)^)	α
**Pt NDs**	41.20	6.837	104.9	0.9
**Pt NCs**	50.21	24.60	69.77	0.914
**Pt/C(20 wt.%)**	95.51	55.67	27.46	0.92

**Table 2 ijms-23-15034-t002:** Analysis of EIS measured in 0.1 M KOH.

Catalyst	Rs (Ω)	Rct (kΩ)	CPE (μF.s^(1−a)^)	α
**Pt NDs**	81.86	18.92	173.5	0.769
**Pt NCs**	89.39	25.17	124.9	0.768
**Pt/C (20 wt.%)**	149.40	52.20	89.39	0.87

**Table 3 ijms-23-15034-t003:** Analysis of EIS measured in 0.5 M NaHCO_3_.

Catalyst	Rs (Ω)	Rct (kΩ)	CPE (μF.s^(1−a)^)	α
**Pt NDs**	87.18	25.65	112.4	0.832
**Pt NCs**	105.9	25.91	108.2	0.789
**Pt/C (20 wt.%)**	160.60	33.28	47.78	0.89

## Data Availability

The data presented in this study are available on request from the corresponding authors. TEM and particle size distribution of commercial Pt/C catalyst, TEM images of catalysts before and after CO oxidation stability, and comparison table for CO oxidation performance can be found in Appendix A.

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
