# Peer review of "Pt-Based Nanostructures for Electrochemical Oxidation of CO: Unveiling the Effect of Shapes and Electrolytes"

_ijms, 2022, doi:10.3390/ijms232315034_

Round 1

Reviewer 1 Report

The authors have studied CO-oxidation of Pt catalysts under different electrolytes, HClO4, NaOH, and Na2CO3 (or NaHCO3, more details shall be provided). This is a fairly complete study; nevertheless, no significant conclusions are made. The work will be more interesting if 1) the discussion on current density and onset potential are based on ECA normalized data; 2) ECSA extracted by CO-UPD and HUPD can be systematically compared.

The EIS measurement shall be analyzed under a certain electrochemical potential or current density, which determine the Rct value. The authors shall specify these details. Also, it is puzzling to the reviewer why the Rs will be different on samples in the same electrolyte and setup. 

There are some errors in references to figures and tables. For example, there are no supporting figures nor tables, but supporting materials are mentioned a few times in the paper. The authors may also want to refer to subfigures together with the number, for example, figure 1i rather than figure i. These inconsistencies/mistakes need to be corrected.

Reviewer 2 Report

Please find the attached.

Round 2

Reviewer 1 Report

The authors have made revisions according to the reviewer's comments. Nevertheless, there are a number of typos and errors the reviewer recommends the authors to correct before publishing the results. The list may not be extensive, and the authors are encouraged to cross-check, pass along to among multiple authors to make sure all are satisfied on the quality of this paper as it will be publicly available.

a)Typos below, for example

line 216 poetical--potential

table S2, typos on '/'?

table S3, NaHCO3 or Na2HCO3?

figure S2, HCLO4--HClO4

b) Tables S1 and S2 both have significant number issues. The number shall be reasonable and consistent.

c) All references in the supporting info shall add journal names.

Reviewer 2 Report

The authors have addressed all concerns raised by the reviewer. The revised manuscript can now be published.
